# Persistent Illegal Hunting of Wildlife in an African Landscape: Insights from a Study in the Luangwa Valley, Zambia

**DOI:** 10.3390/ani14162401

**Published:** 2024-08-19

**Authors:** Paul Zyambo, Jacob Mwitwa, Felix Kanungwe Kalaba, Eustarckio Kazonga

**Affiliations:** 1School of Postgraduate Studies, University of Lusaka, Lusaka P.O. Box 36711, Zambia; 2School of Applied Sciences, Kapasa Makasa University, Chinsali P.O. Box 480195, Zambia; jacob.mwitwa@kmu.ac.zm; 3School of Natural Resources, Copperbelt University, Kitwe P.O. Box 21692, Zambia; kanungwe@cbu.ac.zm; 4School of Medicine and Health Sciences, University of Lusaka, Lusaka P.O. Box 36711, Zambia; ekazonga@yahoo.co.uk

**Keywords:** drivers of illegal hunting, intervention measures, law enforcement, local illegal hunters, Luangwa Valley, persistent illegal hunting, poaching, survival, sustaining livelihoods

## Abstract

**Simple Summary:**

The illegal hunting (poaching) of wildlife has persisted for decades and adversely effected wildlife populations in the Luangwa Valley, Zambia. It has not been clearly understood why illegal hunting has continued notwithstanding efforts to control it. Therefore, this study was conducted to understand why illegal hunting has persisted despite increased mitigation efforts. The study revealed that illegal hunting has continued because its main root causes are the critical need for survival and sustaining the livelihoods of local hunters, which were not adequately addressed. Furthermore, law enforcement, which was the main intervention measure, failed to adequately deter local hunters from poaching wildlife because it could not address the main root causes of illegal hunting. The study concluded that the illegal harvesting of resources in protected areas may persist when the local people’s key motivations for illegal harvesting relate to their critical need to survive and maintain their livelihoods and the main mitigation efforts do not address these critical needs. These study findings are valuable in providing an understanding of how the critical need for survival and maintaining livelihoods and other factors influence the persistence of illegal hunting and in guiding the development of a strategy for the effective control of poaching in the Luangwa Valley.

**Abstract:**

Decades of illegal hunting (poaching) have adversely affected wildlife populations and thereby limited sustainable wildlife conservation in the Luangwa Valley, Zambia. Despite intervention efforts to address the problem, the illegal hunting of wildlife has persisted. Therefore, this study was conducted to understand the persistence of illegal hunting by investigating the drivers of poaching and intervention measures using a mixed methods approach. Stratified random sampling was used to collect data from 346 respondents through structured questionnaires. Purposive sampling was used to collect data through nine focus group discussions and three in-depth interviews with experts. The study revealed that persistent illegal hunting was mainly driven by people’s critical need for survival and sustaining their livelihoods and not by inadequate law enforcement as presumed by resource managers. Although law enforcement was the most prevalent intervention measure, it did not deter local illegal hunters because their main motivations for poaching were not effectively addressed. The key implication of these findings is that where the illegal harvesting of natural resources in protected areas by local resource users is driven by people’s critical need for survival and a livelihood, which is ineffectively addressed, illegal harvesting may persist even with increased law enforcement. This study provides empirical evidence, novel conceptual knowledge and an understanding of how prevalent drivers of poaching and other factors may have influenced persistent illegal hunting in the Luangwa Valley.

## 1. Introduction

The illegal hunting of wildlife is an important contributor to the global problems of biodiversity loss, environmental degradation, climate change and zoonotic pandemics [1,2,3,4,5]. Illegal hunting is generally defined as any extraction of wildlife that is not explicitly authorised by the state or private owners of wildlife [6,7,8] and usually considered synonymous to the poaching of wildlife [9]. Despite huge and increased investments in anti-poaching measures, illegal hunting has persisted and continued to increase, and it is increasingly being recognised that interventions are failing to effectively address the problem [10,11,12,13]. The failure to effectively address illegal hunting has generally been attributed to a poor understanding of illegal hunting and what motivates people to hunt illegally [8,10]. In a recent study, a precursor to the current, Zyambo et al. [14] postulated that the persistence of illegal hunting by local hunters in Africa was associated with the prevalence of drivers of poaching that relate to their need for survival and to sustain their livelihoods and with the prevalence of ineffectively addressed drivers of poaching. They further suggested that the main anti-poaching measures in Africa were mostly designed to address poaching as an activity instead of the drivers of illegal hunting among local hunters. However, these assertions need to be tested in different areas to ensure their validity is based on empirical evidence.

The illegal hunting of wildlife is often conceptualised as a complex phenomenon with various variables that include economic, ecological, cultural, socio-psychological and socio-political perspectives [7,15]. Currently, there is no specific single theoretical underpinning for the illegal hunting phenomenon that is robust enough to accommodate the complexity of the illegal hunting process, despite the contemporary emphasis on instrumental economic theories in the literature. However, there are different theories from different disciplines of science that explain the illegal hunting phenomenon and may vary in their focus depending on prevailing local situations. Based on the wide range and nature of the drivers of illegal hunting identified in Africa [14,16], the theoretical framework which is probably relevant for this study comprises five theories from the behavioural ecology, environmental criminology, socioeconomics, social psychology and socio-political disciplines of science. The first theory that underpins illegal hunting is the Optimal Foraging Theory (OFT), because hunters make decisions in hunting for bushmeat that are usually consistent with the OFT [17,18]. The OFT predicts that the decisions that maximise energy per unit time and thus deliver the highest payoff will be selected for and persist [19]. Similarly, hunters make decisions in hunting for bushmeat or other material parts in accordance with the Rational Choice Theory (RCT), where people are expected to estimate the likely costs and benefits of an action before acting [20]. Both the OFT and RCT focus on the consequences or outcomes of contemplated behaviour. The third theoretical underpinning of illegal hunting is the Situational Precipitator of Crime (SPC) or Situational Precipitator Framework (SPF), where any aspect of the immediate environment may continue to create, trigger or intensify the motivation to commit a crime [21]. Thus, the SPC or SPF focuses on incidents and the stimuli or antecedents of contemplated illegal behaviour such as illegal hunting [22]. The fourth theory is the Defiance Theory (DT), which holds that “sanctions perceived as unfair by way of harsh and disrespectful treatment from the sanctioning agent or by lack of procedural fairness will result in a delegitimisation of authorities and furtherance of crime” [7,23]. The prediction based on the DT is that environmental harm, which includes illegal hunting, will increase (or persist) as the legitimacy of conservation policies, tactics and authority decline [24]. The fifth theoretical underpinning of the illegal hunting phenomenon is the Theory of Planned Behaviour (TPB), which holds that attitude (behavioural beliefs), subjective norms (normative beliefs) and perceived control (control beliefs) are determinants of both intention and behaviour [25,26]. Thus, TPB connects beliefs to intentions and behaviour and implies that behavioural intention is the most immediate determinant of social behaviour [27], which includes illegal hunting. Despite having respective assumptions that are possibly inadequate and may not be always valid for the illegal hunting phenomenon, the collective aforesaid theories provide complementary, broader perspectives and an understanding of complex factors and the processes by which individuals engage and persist in the illegal hunting of wildlife.

In Zambia’s Luangwa Valley, the problem of the illegal hunting of wildlife has persisted for decades despite increased anti-poaching measures such as law enforcement, community-based conservation approaches and investments, and technical and financial support from various stakeholders [28,29,30,31,32]. Consequently, populations of threatened species like elephants (*Loxodonta africana*), black rhinoceros (*Diceros bicornis*), lions (*Panthera leo*), African wild dogs (*Lycaon pictus*) and others have been adversely affected during the last four decades [28,31,32,33,34]. Illegal hunting in the landscape had, by 1995, caused the local extirpation of black rhinoceros from a population of about 4000 in 1973 and a reduction in the elephant population of 75%, to 20,200 animals, between 1970 and 2012 [35,36]. The annual elephant mortalities due to illegal hunting in the Luangwa Valley have remained the highest in the last two decades among wildlife landscapes in the country [32]. Furthermore, other studies in the Luangwa Valley have reported that high levels of illegal hunting via snaring have increased mortality rates and disturbed the population structures of wild animals [29,30,31,37]. Pervasive and persistent illegal hunting via snaring is probably indicative of prevalent local community involvement in poaching, inadequate local community support for conservation and ineffective available measures for addressing poaching via snaring [29,38]. However, it is not clearly established why the illegal hunting of wildlife has persisted despite increased law enforcement and other anti-poaching efforts in the Luangwa Valley.

A few, and mainly not recent, studies in the Luangwa Valley by Leader-Williams et al. [33], Milner-Gulland and Leader-Williams [39], Jackmann and Billiouw [34], Brown and Marks [40], Kings [41] and Nyirenda et al. [32] identified varied and non-comprehensive drivers of poaching, which thereby led to an equivocal understanding of this phenomenon in the landscape. Similarly, Gibson and Marks [28] and Brown and Marks [40] suggested that a poor understanding of the motivations for illegal hunting among local communities was the main reason for intervention measures that were ineffective and inappropriately aimed at poaching activities in the central Luangwa Valley. Therefore, the main objective of this study is to understand why the occurrence of illegal hunting has persisted among local communities despite increased intervention efforts in the Luangwa Valley. Specifically, this study (1) investigates and interprets how the drivers of illegal hunting influence the persistence of illegal hunting by local hunters in the Luangwa Valley; and (2) investigates and interprets how intervention measures affect the drivers and persistence of the illegal hunting of wildlife by local hunters in the Luangwa Valley. Further, this study tests two hypotheses based on the conceptual view and postulation advanced by Zyambo et al. [14] that illegal hunting in Africa is mainly driven by the critical need for survival and sustaining livelihoods and is linked to prevalent and ineffectively addressed drivers of illegal hunting that relate to survival and sustaining livelihoods. Therefore, the first hypothesis (H_1_) is that persistent illegal hunting is associated with the prevalence of drivers of illegal hunting that relate to survival and sustaining the livelihoods of local communities in the Luangwa Valley. The second hypothesis (H_2_) is that persistent illegal hunting is associated with the prevalence of the unsatisfactory performance of intervention measures in addressing the prevalent drivers of illegal hunting in the Luangwa Valley. This study employs an explanatory, sequential, mixed methods design where the findings from the quantitative survey are clarified, confirmed and enhanced, providing a deeper understanding through the consequential use of qualitative study methods.

This represents the first study to establish that persistent illegal hunting in the Luangwa Valley is mainly driven by people’s critical need for survival and sustaining their livelihoods and not by inadequate law enforcement as assumed by resource managers. Further, despite being the major intervention measure, law enforcement failed to deter local illegal hunters because it could not adequately address the main drivers of poaching. This study is also the first to underscore the significance of beliefs, behavioural intentions to hunt illegally and defiance as critical contributory factors driving illegal hunting in the Luangwa Valley. Consequently, a different and novel perspective is advanced for addressing the problem of illegal hunting in the Luangwa Valley by shifting the focus to interventions that are specifically targeted to drivers of illegal hunting instead of symptomatic poaching activities.

## 2. Materials and Methods

### 2.1. Study Area

The study area is located in the Game Management Areas (GMAs) that are adjacent to Luambe, Lukusuzi, North Luangwa and South Luangwa National Parks in the Luangwa Valley in the eastern part of Zambia (Figure 1). GMAs are a category of wildlife-protected area in Zambia in which humans, such as local communities, investors and other stakeholders, are permitted to coexist with wildlife. The GMAs in the Luangwa Valley include Chisomo, Lupande, Lumimba, Mukungule, Munyamadzi, Musalangu, Sandwe and West Petauke.

The Luangwa Valley is characterised by riverine habitats along the Luangwa River, which runs in the southeast direction from the Nyika plateau to where it joins the Zambezi River in the Luangwa District and dissects the escarpment leaving recent alluvium with levees, point bar deposits, flood channels, abandoned channels, oxbow lakes and plains [42]. The landscape is mostly covered by mopane woodland, which is dominated by *Colophospermum mopane*, with other woodlands such as *Combretum*/*Terminalia*, *Acacia*/*Combretum* and *Brachystegia*/*Julbenadia* [43,44]. Its diverse wildlife is utilised through eco-tourism and sustainable hunting. The local communities in the GMAs are involved in the management of wildlife through Community Resource Boards (CRBs), which participate in protecting wildlife and benefit sharing from wildlife utilisation.

### 2.2. Survey Design

The survey design used stratified random sampling for quantitative survey part and stratified purposive sampling for the qualitative survey as the mixed methods approach adopted for this study. Stratified sampling was used in the study because the target population in the Luangwa Valley was expected to be heterogeneous. Stratification into sub-populations was adopted to ensure that representative samples were achieved from sub-populations which were as homogeneous as possible [45,46]. The study population of the Luangwa Valley was therefore stratified into four strata, according to characteristics of the target groups in the area, as (1) Reformed Illegal Hunters, (2) Community Resource Board members, (3) Wildlife Agency Staff and (4) staff members of Conservation-Interested Entities. This was carried out to facilitate the determination of similarities and differences in the experiences, views and perspectives of the four targeted groups on illegal hunting, drivers of illegal hunting and the intervention measures being implemented.

### 2.3. Quantitative Approach

#### 2.3.1. Study Population and Sample Size

The study targeted four sub-groups of wildlife stakeholders from the four strata which were spread in the eight GMAs that straddle nine districts within the Luangwa Valley. The sub-groups from the four strata were considered study targets because they resided within the Luangwa Valley and had a deep understanding of illegal hunting and interventions through direct observations and experiences. The Reformed Illegal Hunters were direct wildlife resource users and were expected to provide credible information on the drivers or motivations for illegal hunting in the Luangwa Valley because they hunted illegally before they surrendered their hunting gear, stopped poaching and were consequently pardoned under an amnesty. Due to that amnesty, they were not afraid to provide sensitive information on the poaching activities they had conducted.

The Conservation-Interested Entities and Wildlife Agency Staff supported and implemented intervention measures in the area, respectively, and could provide appropriate information on the poaching situation and the performance of intervention measures. Wildlife Agency Staff were the official wildlife resource managers in the area. The Conservation-Interested Entities comprised non-governmental conservation organisations, civil societies, government departments such as the forestry and veterinary departments and tourism agencies which supported wildlife conservation through advocacy, funding, the provision of equipment and collaborative efforts. The Community Resource Boards were representatives of local communities in the Luangwa Valley that had an interest in the conservation of wildlife and the wellbeing of communities and would therefore provide information on illegal hunting, conservation measures and community needs.

The total population (N) of the study area was 2078 and comprised four population sub-groups: Reformed Illegal Hunters = 955; Wildlife Agency Staff = 512; members of Community Resource Boards = 367 and staff members of Conservation-Interested Entities = 244. The population information was obtained from the Department of National Parks and Wildlife, Conservation-Interested Entities and tourism businesses in the study area. The population size of the Reformed Illegal Hunters was provided as a cumulative figure of 1069 (from 2007 to 2021) for the valley area only. Consequently, 1069 was corrected with the average adult male mortality rate for the ages between 15 and 50 years, estimated at 7.59 per 1000 for the country, which was calculated by averaging the mortality rates at five-year intervals provided by Zambia Statistics Agency et al. [47]. Therefore, using the average mortality rate to correct for cumulative populations for each year up to 2022 gives 955 as the estimated available population of Reformed Illegal Hunters in the study area. The total sample size (n) of this study was calculated using the Yamane formula (adopted from Yamane [48]), also referred to as the Slovin formula [49], to ensure sample sizes were statistically valid and representative. The calculated total sample size was divided according to the proportion of the population in each sub-group to determine sample sizes for each respective stratum. The study area had heterogenous population, but its degree of variability was unknown. Therefore, the estimated conservative sample size of the study area was assumed to be within a 95% confidence level with a 5% margin of error and the population having a maximum degree of variability (at an estimate population proportion of 0.5). The sample size was calculated as follows:$$n=\frac{N}{1+N\varepsilon 2}$$
where N = population size; n = sample size; and ε = 5% margin of error.

Therefore, the calculated total sample size (n) was 336 (16.2% of the study area population size N). Each stratum contributed proportionately to the total sample size as Reformed Illegal Hunters = 150; Wildlife Agency Staff = 84; Community Resource Boards = 61; and Conservation-Interested Entities = 41.

#### 2.3.2. Data Collection Methods and Tools

Data collection methods for the quantitative surveys in the four strata included cluster random sampling and stratified random sampling. The cluster random sampling technique was used to survey in the Reformed Illegal Hunters stratum from chiefdoms in the study area. The 18 chiefdoms in the study area were considered clusters and available Reformed Illegal Hunters in the randomly selected clusters were given opportunities to respond to the questionnaire administered by the researcher. The cluster random sampling in the Reformed Illegal Hunters stratum made sampling easier and more efficient as respondents were widely dispersed. Stratified random sampling was used to survey the Community Resource Boards, Conservation-Interested Entities and Wildlife Agency Staff strata, as respondents were easily found after being randomly selected from respective lists using computer-generated random numbers. Data collection for the quantitative survey and subsequent qualitative survey was conducted from March 2021 to October 2022.

##### Questionnaire

The questionnaire was designed to collect data that were focused on respondents’ experiences, views and perspectives on the levels, trends and persistence of illegal hunting; drivers of illegal hunting; intervention measures implemented; and the impacts of intervention measures in addressing the drivers of illegal hunting (see Appendix A). The questionnaire was structured with options for selecting applicable responses and, where possible, respondents could include other views. A structured questionnaire was useful as it facilitated the collection of quantitative data to be compared between respondent groups but might have limited the expression of different views and experiences by respondents [46]. The use of a qualitative approach in this study helped capture different views and experiences that were not collected in the quantitative approach through a questionnaire. Another challenge in using the structured questionnaire was that some participants, especially Reformed Illegal Hunters, were unable to read or write. This was addressed by having the researcher read, in their respective local language, translations and record their responses on the questionnaires accordingly.

The questionnaire had four sections or constructs designed for data collection that addressed specific study objectives. The questions in the first section were aimed at collecting personal data such as the participants’ age group, gender/sex, wealth status, sources and levels of income and educational level. The second section targeted the collection of data on participants’ observations, experiences and views on the levels, trends and persistence of poaching activities in the study area. The section on drivers/motivators for poaching required participants to choose any from a list of 23 drivers or motivations for illegal hunting that prevailed in the study area. To ensure the list of drivers of illegal hunting was as exhaustive as possible, the list was adopted from the published literature [16,50]. In the fourth section, participants were asked to choose intervention measures that were employed to deal with the drivers of poaching selected in the previous section. Participants were asked to grade the performance of the intervention measures implemented to tackle the selected drivers of illegal hunting. Further, the participants were also asked to indicate their preferred intervention measures to be implemented in the study area to deal with the drivers of illegal hunting.

### 2.4. Qualitative Approach

The qualitative data collection from the sampling strata was achieved by purposively selecting participants’ peers for focus group discussions (FGDs). The numbers of the focus groups were determined by the issues from the quantitative survey that required clarification, conformation or a deeper understanding. The participants, during in-depth interviews (IDIs), were also purposively sampled based on the adequacy of their individual expertise and vast experience of at least 10 years in wildlife conservation in the Luangwa Valley.

#### 2.4.1. Focus Group Discussions

The FGDs were held in participants’ respective strata to collect qualitative data to confirm, clarify and provide a deeper understanding on some aspects identified during quantitative data collection. The FGDs used a semi-structured guide to ensure the discussions focused on the levels, trends and drivers of illegal hunting and intervention measures implemented in the area and their impacts in addressing drivers (see Appendix A). The objectives for holding FGDs in this study was to qualitatively obtain detailed explanations and understandings among peers in the respective strata on the illegal hunting phenomenon and create reference groups for confirming and clarifying the issues that emerged from the initial quantitative data collection and analysis. The FGDs were also used to collect data on behavioural intentions to hunt illegally and local beliefs about illegal hunting. Local beliefs about illegal hunting included aspects such as whether poaching was acceptable in some situations (behavioural beliefs), whether immediate families and communities would accept, support or facilitate poaching in some situations (normative beliefs) and whether individuals themselves had the power to start or stop poaching (control beliefs).

Each FGD was held with about 10 people who were peers with similar social power and characteristics. Matthews and Ross [46] suggested that focus groups should have between 5 and 13 members who have something in common with the topic to ensure effective facilitation. Nyumba et al. [51] also found that the median number of participants in focus groups in 170 reviewed published papers on conservation was 10. Thus, the researcher guided each FGD comprising about 10 people with a structured guide on thematic areas of the study objectives. The FGDs were flexible and concentrated on themes that had a lot of issues to discuss and clarify. The researcher took notes and recorded the proceedings.

#### 2.4.2. In-Depth Interviews

The IDIs were conducted with three purposively selected experts who had at least 10 years of experience in matters relating to illegal hunting, intervention measures and the conservation of wildlife in the Luangwa Valley and were able to share their experiences, views and perceptions. Three experts were purposively selected to collect their respective perspectives from the implementors of conservation strategies that support law enforcement and sustainable community livelihoods. The objective of holding IDIs was to provide further understanding on the persistent illegal hunting in the Luangwa Valley relating to the levels, status and persistence of illegal hunting and the performance of intervention measures. The researcher conducted IDIs in accordance with the guide (see Appendix A) and recorded the proceedings accordingly.

### 2.5. Validity and Reliability of Study Instruments

The content validity of the questionnaire, which is the degree to which the questionnaire measures what it was intended to measure [46], was enhanced by piloting or pre-testing it on 31 respondents (9.2% of the study sample size) to identify validity issues and other problems relating to questions, format and scale. The validity of each questionnaire item was tested using Pearson’s correlation coefficients during the pilot or pre-test survey, where responses to each questionnaire item correlate with the overall responses. Furthermore, a triangulation of methods (quantitative and qualitative) was used to confirm the validity and credibility of the study tools and findings. To ensure that the reliability or the ability of the questionnaire to reproduce consistent results if the study was repeated or replicated using the same questionnaire [52], this was checked during the pilot or pre-test survey. The internal consistency of the study tool was tested using the coefficient of Cronbach’s Alpha as suggested by Heale and Twycross [53].

The trustworthiness of qualitative aspects of this study was concerned with the truth value of the qualitative data, analyses and interpretations [54] from focus group discussions and in-depth interviews. The criteria for enhancing trustworthiness, as proposed by Lincoln and Guba [55] and Guba and Lincoln [56] and elaborated on by Cope [57] and Nowell et al. [58], was adopted. Particularly, the researcher (first author) prolonged their engagement with the participants and conducted many visits to the study area over six months during data collection. The researcher followed up with the participants through a member checking process with questions to clarify and confirm the data and the interpretation of the data. The researcher conducted peer and expert debriefing where colleagues and experts with related experience in the study subject matter looked at the data and verified their interpretation. The triangulation of data collection and these methods was achieved using questionnaire surveys, focus discussion groups, in-depth interviews and a review of the literature. Credibility was also enhanced by conducting a negative-case analysis and reporting accordingly when opposing and different data were identified in the study. The researchers also endeavoured to express participants’ feelings and emotions correctly by including quotes from participants in their descriptive approach. The interpretations and conclusions were drawn directly from the data and from the congruence of two or more FGDs and IDIs on the data’s accuracy, relevance and meaning.

### 2.6. Data Analyses

Data analyses were performed to address the study objectives by generating descriptive and inferential statistics such as frequency distribution statistics, the comparison of the frequency distributions of variables between strata, tests of the associations of variables, hypotheses testing and the prediction of a high persistence of illegal hunting. For qualitative data, the audio recordings of the FGDs and IDIs were transcribed. Related items were coded for each transcription, and themes were generated from the grouping of codes. Thematic data analysis was conducted by using NVivo 12 software. The qualitative data analysis generated themes and sub-themes relating to the drivers of illegal hunting, intervention measures, law enforcement, local hunters’ beliefs and behavioural intentions to hunt illegally, as expressed in FGDs and IDIs. The total numbers of references made to the themes or sub-themes by participants in the FGDs and IDIs indicated the prominence of these themes and sub-themes. The number of FGDs and IDIs on particular themes and sub-themes the indicate congruence level of those themes and sub-themes.

#### 2.6.1. Response Frequencies of Drivers of Illegal Hunting and Intervention Measures

The number of responses confirming occurrences of the drivers of illegal hunting, intervention measures, the most satisfactory performance rating of intervention measures and unsatisfactory performance rating of intervention measures in the study area were calculated as percentages of the total responses for each variable. These frequencies indicated the prevalence level of these respective variables. The drivers of illegal hunting were further categorised as proximate, underlying or thematic, as adapted from Geist and Lambin [59] and Jellason et al. [60] and redefined by Zyambo et al. [14]. Accordingly, proximate drivers of illegal hunting were considered any immediate desires, feelings, lack or needs of humans at a local level that directly induced them to hunt illegally. Likewise, underlying drivers were defined as factors that underpin, enhance or enable proximate drivers and might work at the local level or have an indirect influence from the national or global levels. The thematic drivers of illegal hunting referred to a category that best described the fundamental characteristics or attributes of either proximate or underlying drivers.

#### 2.6.2. Relationships between Drivers of Illegal Hunting and Intervention Measures

The frequencies of drivers of illegal hunting and the frequencies of their respective intervention measures were used to test the relationship between the prevalence of drivers of illegal hunting and the prevalence of intervention measures using Spearman’s rank correlation coefficient. Similarly, correlations between the prevalence of drivers of illegal hunting and the most satisfactory performance rating of respective intervention measures and between the prevalence of drivers of illegal hunting and unsatisfactory performance ratings of their respective intervention measures were tested using Spearman’s rank coefficients. The statistical tests were carried out with IBM SPSS Statistics version 27 software and considered *p*-values less than 0.05 as statistically significant.

#### 2.6.3. Comparing Responses in Sampling Strata

To ascertain the differences in responses in the sampling strata, the proportional means of the positive responses on prevalent drivers of illegal hunting in each stratum were compared and tested for differences using an ANOVA F-Test for Proportions. A significant test among the sampling strata required post hoc comparisons with a Tukey HSD Test to identify significant pairwise differences among the groups’ proportional means. The response differences in the sampling strata were further determined by comparing the distribution of responses on the prevalent drivers of illegal hunting in the strata with the distribution of strata population sizes. The Goodness of Fit test was used to compare these distributions. The statistical tests were carried out with IBM SPSS Statistics version 27 software and considered *p*-values less than 0.05 as statistically significant.

#### 2.6.4. Hypotheses Testing

Hypotheses testing was performed by testing the null hypotheses using statistical tests of independence (or association) between the persistence of illegal hunting and the prevalence of drivers of illegal hunting and the prevalence of the unsatisfactory performance of intervention measures. The association test procedure results in likelihood ratios (and χ^2^ values). The calculated *p*-values of the likelihood ratios or χ^2^ values for the associations were compared with the significance level (∝) to determine the statistical significance of the associations. If the *p*-value < ∝, the null hypothesis was rejected and the alternative hypothesis was accepted and if the *p*-value was > or = ∝, the null hypothesis was accepted and the alternative hypothesis was rejected. The significance level (∝) for the rejection and acceptance of the null hypotheses was taken as 0.05 for two-tailed tests of the two hypotheses.

**Hypothesis H1_A_:** 
*The persistent occurrence of illegal hunting in the Luangwa Valley is not associated with the prevalence of drivers of the illegal hunting of wildlife that relate to the survival and livelihoods of local illegal hunters.*


**Hypothesis H1_B_:** 
*The persistent occurrence of illegal hunting in the Luangwa Valley is associated with the prevalence of drivers of the illegal hunting of wildlife that relate to the survival and livelihoods of local illegal hunters.*


**Hypothesis H2_A_:** 
*The persistent occurrence of illegal hunting is not associated with the prevalence of the unsatisfactory performance of intervention measures addressing the drivers of illegal hunting of wildlife that relate to the survival and livelihood of local illegal hunters.*


**Hypothesis H2_B_:** 
*The persistent occurrence of illegal hunting in the Luangwa Valley is associated with the prevalence of the unsatisfactory performance of intervention measures addressing the drivers of illegal hunting of wildlife that relate to the survival and livelihood of local illegal hunters.*


### 2.7. Ethical Considerations

The ethical issues in this study are considered critical to the successful completion of the study. A consideration of ethical issues during the study was important to protect research participants, develop trust with participants, enhance the integrity of the research and avoid misconduct and impropriety [50]. Initially, the researcher sought clearance from the Research Ethics Committee of the School of Postgraduate Studies at the University of Lusaka before commencing data collection (see Appendix A). The researcher also sought permission from relevant institutions which were directly responsible for the area; in this case, the Department of National Parks and Wildlife. Furthermore, during data collection, the researcher sought consent from all participants before they responded to questionnaires or participated in the interviews or discussions and before they had their proceedings recorded. Participants in the study expressed their consent to participate by signing an Informed Consent Form for participants (see Appendix A). Informed consent involved consenting to participate after being informed of the identity of the researcher, the purpose of the study, the benefits and risks of the study for participants, the anonymity of participants, a guarantee of confidentiality to participants and assurance that participants could withdraw from participating at any time they felt necessary.

## 3. Results

### 3.1. Validity, Reliability and Trustworthiness of Study Instruments and Process

The results from pre-testing the questionnaire showed that few items had responses with non-significant Pearson’s correlation coefficients and problems which respondents found it difficult to respond to. Conversely, most items were not wrongly responded to, the responses were not mutually contradictory, and the coefficient of Cronbach’s Alpha was 0.9 (n = 86). The questionnaire items which were difficult to respond to or had non-significant Pearson’s correlation coefficients were corrected or left out before the study’s data collection began. The internal consistency of the corrected questionnaire after the study’s data collection showed the coefficient of Cronbach’s Alpha as 0.8 (n = 68).

The trustworthiness of the qualitative data’s collection, analysis and interpretation was enhanced by engaging participants for over six months to administer questionnaires, hold FGDs and IDIs and conduct member checking to clarify and confirm the data collected and its understanding and interpretation by the researcher. Recordings of nine FGDs and three IDIs, with their respective transcripts, were made and secured for reference, analysis and independent audit. The results from the thematic data analysis indicate that congruence was attained in 87.3% of the combined themes and subthemes between at least two or more FGDs or IDIs and between FGDs and IDIs. The FGDs and IDIs confirmed most results from the quantitative survey on the prevalent drivers of illegal hunting and prevalent intervention measures and provided clarification or new understandings of law enforcement, defiance, local beliefs about the illegal hunting of wildlife and behavioural intentions to hunt illegally. Their interpretation was based on the congruence and confirmability of the data generated by quantitative, qualitative and literature review approaches. Data with negative, opposing or different perspectives were analysed, reported and considered in the interpretation. During discussions held by the researcher on this study’s qualitative data collection and interpretation, peers and experts indicated that the collected data and resulting interpretation were inclusive, remarkably insightful and most likely representative of the reality of persistent illegal hunting in the Luangwa Valley.

### 3.2. Quantitative Approach

#### 3.2.1. Demographics and Socio-Economic Characteristics of Respondents

A total of 346 respondents were sampled from four strata, representing 16.6% of the total study population of 2078 in the Luangwa Valley. The respective stratified sample sizes were based on the proportion of the population in each stratum. The distribution of the stratified collected samples (142, 94, 65 and 45) was not significantly different from the distribution of the sub-populations in the strata *(χ*^2^ = 3.437, *df* = 3, *p* = 0.329). Most of the respondents were male (n = 310, 89.6%) and 52.3% (n = 181) of respondents considered themselves poor. A total of 174 (50.3%) respondents indicated the hunting of wildlife (n = 122, 35.3%) and farming (n = 52, 15.0%) as their major non-employment-based sources of income. However, 150 (43.4%) respondents specified employment as their major source of income. The age of most of respondents (n = 219, 63.3%) ranged from 30 to 50 years old. Few (n = 37, 10.7%) were below 30 years old and 89 respondents (27.7%) were older than 50 years old. A total of 174 (50.3%) respondents had not completed senior secondary education, with 60.3% of these (n = 105) having only attained a primary school education level.

#### 3.2.2. Illegal Hunting’s Drivers, Levels, Trends and Persistence

Table 1 shows the results of the status of illegal hunting in the Luangwa Valley. A total of 189 (54.6%) respondents indicated that the illegal hunting level was moderate to high, with 78 (22.5%) indicating a moderate to common use of firearms/snares as evidence of the high illegal hunting level in the Luangwa Valley. Over 50% (n = 184, 53.2%) of respondents indicated that illegal hunting had persisted for up to 14 years (a short time), whereas a total of 153 (44.2%) indicated a persistence period of 15 to over 30 years (a long to very long time) in the landscape. Responses on illegal hunting’s persistence correlated positively with the age groups of the respondents (*r_s_* = 0.108, *df* = 343, *p* = 0.046). Respondents indicated that illegal hunting had persisted in the Luangwa Valley because of the prevalent drivers of illegal hunting (n = 70, 20.2%) and ineffective intervention measures (n = 83, 24.0%). However, 60.7% (n = 210) of respondents indicated that illegal hunting trends were decreasing with evidence of a declining use of firearms/wire snares in illegal hunting (n = 79), decreasing numbers of people being arrested for illegal hunting (n = 52) and increasing populations of some wildlife species (n = 45).

Table 2 shows the identified drivers of illegal hunting in the Luangwa Valley, categorised as either proximate or underlying and as thematic drivers. Twenty-three drivers of illegal hunting were identified through a questionnaire survey in the Luangwa Valley. The eight most prevalent drivers of illegal hunting in the Luangwa Valley were the lack of alternative sources of income/employment (n = 197, 56.9%), poverty (n = 195, 56.4%), the need for bushmeat consumption (n = 183, 52.9%), the need for income from bushmeat and animal products (n = 180, 52.0%), sponsorship to hunt illegally (n = 132, 38.2%), a lack of sources of meat/protein (n = 112, 32.4%), retaliatory killing (n = 96, 27.7%) and preventative killing (n = 84, 24.3%). Remarkably, among these eight most prevalent drivers, six were proximate drivers, with five of these falling under the thematic driver categorised as the need for survival and sustaining livelihoods. The moderately prevalent drivers of illegal hunting included human–wildlife conflicts (n = 79, 22.9%), the demand for wildlife products (n = 70, 20.2%), inadequate conservation education/awareness (n = 70, 20.2%), inadequate tangible benefits of conservation (n = 52, 15.0%), human population influx/increase (n = 49, 14.2%) and inadequate community involvement in wildlife management (n = 47, 13.6%) and these were categorised as underlying drivers. Surprisingly, weak/inadequate law enforcement (n = 39, 11.3%) was among the nine least prevalent drivers of illegal hunting in the Luangwa Valley, together with cultural/traditional needs (n = 20, 5.9%) and defiance/protesting unfairness (n = 10, 2.9%).

In summary, these results showed that the most prevalent drivers of illegal hunting were mostly proximate, related to people’s needs for survival and sustaining their livelihoods, which contributed to over 30 years of persistence and moderate-to-high levels of illegal hunting in the Luangwa Valley. Further, the results remarkably showed that weak/inadequate law enforcement was among the least prevalent drivers of illegal hunting identified by respondents in the study.

#### 3.2.3. Relationship between Illegal Hunting Drivers and Intervention Measures

Table 3 shows the prevalence (indicated as percent frequency) of intervention measures in addressing the drivers of illegal hunting in the Luangwa Valley. Respondents in a questionnaire survey identified 11 intervention measures that were being implemented in the Luangwa Valley. The most prevalent intervention measures for addressing the drivers of illegal hunting were improving law enforcement (n = 213, 61.6%), conservation education/awareness (n = 207, 59.8%), the provision of alternative livelihoods (n = 187, 54.0%) and the provision of alternative sources of income/employment (n = 152, 43.9%). Among the most prevalent, the first two intervention measures addressed inadequate law enforcement and inadequate conservation education/awareness and were under the category of underlying drivers of illegal hunting, whereas the last two tackled proximate drivers of illegal hunting such as a lack of alternative livelihoods and lack of alternative sources of income/employment. The moderately prevalent intervention measures included community involvement in wildlife management (n = 112, 32.4%), protecting communities from animal attacks and threats (n = 99, 28.6%) and revenue sharing from hunting (n = 80, 23.1%). The leading intervention measure among the moderately prevalent intervention measures addressed the inadequate community involvement in wildlife management, which was categorised as an underlying driver of illegal hunting. The second moderately prevalent intervention measure dealt with both underlying and proximate drivers of illegal hunting such as human–wildlife conflicts and preventative and retaliatory killings. The third moderately prevalent intervention measure addressed the lack of tangible benefits from conservation as an underlying driver of illegal hunting. Further, the intervention measures with the least prevalence included land use planning (n = 62, 17.9%), the provision of bushmeat from hunting (n = 49, 14.2%), the provision of an alternative to bushmeat (n = 27, 7.8%) and the provision of access to wild resources (n = 26, 7.5%). Three of the four least prevalent intervention measures addressed both underlying and mostly proximate drivers of illegal hunting such as the inadequate tangible benefits of conservation, a lack of sources of meat/protein and the need for bushmeat consumption.

Figure 2 indicates the perceived performances of intervention measures (unsatisfactory or the most satisfactory) in addressing the drivers of illegal hunting in the Luangwa Valley. The intervention measures with the most satisfactory performance in addressing the drivers of illegal hunting included law enforcement (n = 130, 48.9%), conservation education/awareness (n = 98, 41.0%) and community involvement in wildlife management (n = 49, 32.0%) and were categorised as underlying drivers. Conversely, intervention measures with predominantly unsatisfactory performances in addressing the drivers of illegal hunting were the provision of alternative sources of income/employment (n = 136, 53.3%), the provision of alternative livelihoods (n = 129, 48.5%), land use planning (n = 72, 52.9%) and the protection of communities from wildlife attacks/threats (n = 60, 32.4%). These predominantly unsatisfactory intervention performances were identified as intervention measures that mainly addressed the proximate drivers of illegal hunting.

The prevalence of the drivers of illegal hunting showed a negative but not significant correlation with the prevalence of their respective intervention measures (*r_s_* = −0.24, *df* = 9, *p* = 0.485). The prevalence of the drivers of illegal hunting was negatively correlated with the prevalence of the respective intervention measures with the most satisfactory performances (*r_s_* = −0.81, *df* = 9, *p* = 0.003). Conversely, the prevalence of the drivers of illegal hunting was positively correlated with the prevalence of the respective intervention measures with unsatisfactory performances (*r_s_* = 0.62, *df* = 9, *p* = 0.040).

#### 3.2.4. Comparisons of Responses in the Strata on the Prevalent Drivers of Illegal Hunting

The proportional means of the responses in identifying prevalent drivers of illegal hunting were significantly different among the sampling strata (*F*
_(3,31)_ = 5.838, *η*^2^ = 0.645, *p* = 0.003). Post hoc comparisons using the Tukey HSD Test showed significant pairwise differences among the group proportional means between the Reformed Illegal Hunters and the Community Resource Boards (*p* = 0.002) and between the Reformed Illegal Hunters and the Wildlife Agency Staff (*p* = 0.025).

Similarly, the distributions of the responses that identified the prevalent drivers of illegal hunting in the Luangwa Valley were significantly different, when tested for Goodness of Fit, to the distribution of the strata population sizes. Responses from Reformed Illegal Hunters were significantly more frequent than expected, with the largest residuals (differences between expected and observed responses) in identifying the following drivers of illegal hunting: the need for bushmeat consumption (*χ*^2^ = 53.598, *df* = 3, *p* < 0.001), the need for income from bushmeat (*χ*^2^ = 40.041, *df* = 3, *p* < 0.001), a lack of alternative sources of meat (*χ*^2^ = 20.741, *df* = 3, *p* < 0.001), preventative killing (*χ*^2^ = 33.742, *df* = 3, *p* < 0.001), retaliatory killing *(χ*^2^ = 19.841, *df* = 3, *p* < 0.001) and sponsorship to hunt illegally (*χ*^2^ = 25.368, *df* = 3, *p* < 0.001). These were identified in significantly more of the responses from Reformed Illegal Hunters and were all among the eight most prevalent drivers. Five of these were classified as proximate drivers and thematically fell under the need for survival and sustaining livelihoods (see Table 2). Responses from Wildlife Agency Staff were significantly more numerous than expected in identifying the following drivers of the illegal hunting: a lack of alternative income/employment (*χ*^2^ = 18.640, *df* = 3, *p* < 0.001), high market demand for wildlife products (*χ*^2^ = 12.079, *df* = 3, *p* = 0.007), a lack of conservation education/awareness (*χ*^2^ = 77.985, *df* = 3, *p* = 0.001), human population increase/influx (*χ*^2^ = 87.779, *df* = 3, *p* < 0.001) and weak/inadequate law enforcement (*χ*^2^ = 37.282, *df* = 3, *p* < 0.001). Four of these drivers of illegal hunting were identified by Wildlife Agency Staff in significantly more than the expected responses, were categorised as underlying drivers and were not categorised under the thematic driver of the need for survival and sustaining livelihoods. The responses from Conservation-Interested Entities were also significantly more prevalent than expected in identifying similar drivers of illegal hunting as those identified as by Wildlife Agency Staff. However, the differences between the expected and observed responses from Wildlife Agency Staff were larger than those from Conservation-Interested Entities.

#### 3.2.5. Hypotheses Testing

Table 4 shows significant associations (*p* < 0.05) between the persistence of illegal hunting and the need for bushmeat consumption, the need for income from bushmeat and other wildlife products, preventative killing, human–wildlife conflicts, the need for trophies for income/use, and the lack of tangible benefits from wildlife conservation. These were directly related to people’s need for survival and to sustain their livelihoods. Except for the need for trophies for income/use, the drivers of illegal hunting that were significantly associated with the persistence of illegal hunting were among the 12 most prevalent drivers of illegal hunting identified in a questionnaire survey (Table 2). Surprisingly, poverty, a lack of alternative income/employment, a lack of alternative sources of meat and retaliatory killing were not significantly associated with the persistence of illegal hunting (*p* > 0.05).

Similarly, Table 5 shows moderate to strong significant associations (*p* < 0.05) between the persistence of illegal hunting and five intervention measures with unsatisfactory performances in addressing the drivers of illegal hunting that directly relate to people’s need for survival and sustaining livelihoods. These intervention measures were addressing drivers of illegal hunting identified as being among the 12 most prevalent in the Luangwa Valley. Surprisingly, unsatisfactory performances in three intervention measures—the provision of alternative employment/income and the protection of communities from threats and attacks from wild animals—were not significantly associated with the persistence of illegal hunting in the Luangwa Valley (*p* > 0.05).

### 3.3. Qualitative Approach

A qualitative approach was adopted to confirm and clarify the results obtained through the quantitative approach and to provide a deeper understanding of the illegal hunting phenomenon in the Luangwa Valley. Nine FGDs with total of 93 participants and three individual IDIs were held. The results showed that nine broad themes emerged from thematic data analysis (Appendix A).

#### 3.3.1. Drivers of Illegal Hunting in the Luangwa Valley

The results in Appendix A showed 17 subthemes and 1 theme which represented the drivers of illegal hunting discussed in the FGDs and IDIs. The total 18 drivers of illegal hunting included 14 drivers of illegal hunting similar to those identified by the quantitative approach and 4 others exclusively identified by the qualitative approach (see Appendix A). The drivers of illegal hunting which were exclusively identified by the qualitative approach included poor partnerships/collaborations, human encroachment and development, the non-ownership of wildlife by communities and behavioural intentions to hunt illegally. The results also showed that the prominent drivers of illegal hunting with at least nine cited references included defiance/protesting unfairness, behavioural intentions to hunt illegally, poverty, inadequate law enforcement, the need for income from bushmeat, human–wildlife conflicts, a lack of alternative livelihoods, the limited tangible benefits from conservation and a lack of employment. Six of the nine prominent drivers of illegal hunting were also among the twelve most prevalent drivers of illegal hunting identified by the quantitative approach in this study (see Table 2). Surprisingly, despite being among the least prevalent in the quantitative approach, defiance or protesting unfairness and inadequate law enforcement were among the most prominent, with total 19 and 23 references, respectively, in the FGDs and IDIs. The prominence of defiance or protesting unfairness was identified, with total of 19 references from five of the nine FGDs. However, the prominence of inadequate law enforcement was identified, with 23 references from only one FGD (Wildlife Agency Staff) and three participants in the IDIs. The results showed that the prominence of defiance or protesting unfairness was more widespread among the FGDs than that of inadequate law enforcement, as this perspective was restricted to Wildlife Agency Staff and Conservation-Interested Entities.

The participants in FGDs highlighted defiance/protest unfairness as a driver of illegal hunting and the reasons for defiance through hunting illegally. Six reasons for defiance were given by participants in FGDs and these included 1. unfair responses when the wildlife agency acted more swiftly when an elephant or any species of wildlife was illegally killed than when a local person was killed or injured by a wild animal; 2. the non-availability of compensation when wildlife killed people or no mitigatory action taken to address human–wildlife conflicts; 3. local community people were arrested for illegal hunting for bushmeat whereas Wildlife Agency Staff were tolerated when they hunted illegally; 4. non-local people were employed to work in conservation in preference to local people; 5. revenue generated from wildlife in the local area was used for development in other areas of the country; and 6. limited or no hunting licences for legalised bushmeat were made available to local community members. The following is an example of one of the reasons for defiance as expressed by a FGD participant:

“*For instance, you will find that the elephant has killed someone, and then you hear that there is no compensation”. …. “So due to frustration, they will go and kill the animal”. …” they will kill the animal, and just leave it because of frustration*”, participant #7, Kakumbi CRB FGD.

#### 3.3.2. Limitations of Law Enforcement in Addressing Illegal Hunting

The results in Appendix A showed that the limitation of law enforcement theme was prominent, with a total of 24 references from six FGDs and three IDI participants. The remarkable prominence of the limitation of law enforcement was highlighted by participants in the FGDs and IDIs in response to whether improving law enforcement would deter illegal hunters or ultimately control illegal hunting effectively. The FGD and IDI participants indicated that law enforcement could not deter local illegal hunters or control illegal hunting effectively because the critical motivations for illegal hunting were still prevailing among communities, such as poverty, the need for income, and a lack of livelihoods.

“*So, no matter how many law enforcement scouts there will be, but if the poacher has got no other means, and there is no other way of getting him out of poaching, then poaching will not end to say the truth”. … “They go for poaching due to having nothing to do. So, when they find what to do, they will stop poaching*”, Reformed Illegal Hunter #3, Nyalugwe FGD.

One Reformed Illegal Hunter demonstrated how he could not be deterred from illegal hunting even with the risk of being arrested by wildlife law enforcement staff, as shown by the frequency of him being arrested for illegal hunting: “*I had been arrested seven (7) times and at one time they (wildlife law enforcement officers) even shot me in my leg for poaching*”, Reformed Illegal Hunter #6, Sandwe FGD.

The participants in IDIs also indicated that law enforcement was limited in addressing illegal hunting because it was ineffective, inefficient, insufficient and only evoked negative reactions from illegal hunters.

“*And you know I think another factor is that when you fight them (illegal hunters), as the law enforcement tends to do, they get smarter. They don’t stop, they just get smarter. They know how to hide, they know where to move around, they figure out where to go hunting so they minimize their risk or how to, you know do things in a cleverer way. So, I don’t think fighting them necessarily can reduce poaching, but the problem is when you are unable to sustain your law enforcement, those people will be there waiting and then they will come back with a greater vengeance, with a greater aptitude for poaching*”.

“*… wildlife law enforcement is important, I just don’t think it is efficient…” and “… law enforcement is necessary but it’s not sufficient*”, expert participant #1, IDI.

#### 3.3.3. Unsatisfactory Performance of Intervention Measures

The results in Appendix A showed that under the theme unsatisfactory intervention performance there were five prominent subthemes that had at least six references each and these subthemes included the provision of alternative livelihoods, the provision of employment, improving law enforcement, mitigating human–wildlife conflicts and the provision of tangible benefits from conservation. Except for improving law enforcement, these subthemes were also among the seven most prevalent intervention measures with unsatisfactory performances identified using the quantitative survey approach and reported earlier in this study. The results from the qualitative survey approach were similar to those of the quantitative survey in terms of the unsatisfactory performance ratings of intervention measures.

#### 3.3.4. Beliefs and Behavioural Intentions to Hunt Illegally

The qualitative data analysis results in Appendix A show the behavioural (attitudes), normative and control beliefs that participants in FGDs had about wildlife and illegal hunting and its associated benefits, with an overall total of 63 references. Specifically, behavioural beliefs were the most prominent, with 30 references, followed by control beliefs with 21 references and normative beliefs with 12 references. The subthemes under behavioural beliefs depicted the belief that wildlife was God’s creation given to people for use, food, wealth, survival and livelihood support. The subthemes also showed the belief that although illegal hunting was bad in some ways, it was good, the most critical option for people’s survival and part of their tradition. The following are examples of behavioural beliefs about wildlife and illegal hunting as expressed by participants in the FGDs:

“*Let’s talk about the creation, where you asked us that why did God give us wildlife. Then there were answers that God gave wildlife to man so that he can help himself. So, if I have food, then my neighbour, not just my neighbour even my grandparents have no food not even tea, they cannot even manage to go and work on the farm. Then because of the animals that God has given us, I get up and go and kill one Common duiker and sell for (or barter with) three tins (of grain). I get one tin (of grain) and give them so that they are saved from hunger that means I have saved their lives from dying from hunger*”, Reformed Illegal Hunter #5, Nyalugwe FGD.

The subthemes under the normative beliefs showed that illegal hunting was broadly believed to be helpful to communities and the families of illegal hunters for their survival and livelihood. The subthemes also showed the belief that, despite being not supported by some people, illegal hunting was mostly supported and facilitated by communities and the families of illegal hunters.

“*When the game guard is in a certain area, one just waits for two days as the game guard will move out”. … “That is when you come out and poach from the area he has moved from”. … “Sometimes you get information from people that game guards are in this area, so you go the other way to poach animals*”, Reformed Illegal Hunter #2, Luembe FGD.

In response to whether family members supported and facilitated the preparation for illegal hunting, the participant said, “*yes, it’s just tradition. So, if your wife prepares you well and sees you off then it is well. It means you will go far and well as you go poaching*”, Reformed Illegal Hunter #5, Luembe FGD.

The subthemes on control beliefs showed the common belief that it is easy to start and difficult to stop engaging in illegal hunting. It was less commonly believed that stopping illegal hunting is easy. Participants highlighted these beliefs in the FGDs, giving as the reason for the difficulty in stopping illegal hunting not finding alternatives to illegal hunting.

“*To stop poaching is difficult”. … “Yes, for you to stop poaching you have to find what to do in place of poaching”. … “Starting to poach is easier than stopping. One time a friend of mine poached and gave me some bushmeat and I didn’t have a firearm. The bushmeat was good so I also looked for a firearm and started poaching*”, Reformed Illegal Hunter #8, Mwape FGD.

During the member checking process to confirm whether what was understood by the researcher from the FGDs was correct, that stopping illegal hunting was difficult and that law enforcement would not end poaching in the Luangwa Valley, one reformed illegal hunter in Chief Nyalugwe’s area confirmed that stopping illegal hunting was difficult. He also confirmed that increased law enforcement would not completely address illegal hunting in the Luangwa Valley. He gave a personal example that, even as a reformed illegal hunter, he sometimes hunted illegally when he was hard-pressed due to a lack of food and income for use in the family. He further indicated that engaging in alternative activities that could give him adequate food and income would completely keep him away from poaching wildlife.

The results in Appendix A also showed the occurrence of behavioural intentions to hunt illegally among participants, and this was prominent, with 17 references in six FGDs featuring mainly Reformed Illegal Hunters. Remarkably, in each cited reference for the theme, behavioural intentions to hunt illegally, there were features of the beliefs (behavioural, normative or control) expressed by participants in the FGDs on wildlife, illegal hunting or the benefits derived thereof. This showed the linkage between behavioural intentions to hunt illegally and the beliefs of participants in the FGDs.

“*What causes poaching are the problems that we face in our homes. That is why we go poaching. We kill animals to get help in our homes*”, Reformed Illegal Hunter #1, Jumbe FGD.

## 4. Discussion

This study was undertaken to understand why illegal hunting has persisted in the Luangwa Valley despite various intervention measures being implemented. The study intended to achieve its main objective by determining, analysing and interpreting how drivers of illegal hunting and intervention measures have affected the persistence of illegal hunting in the Luangwa Valley. Here, the study provides empirical evidence that illegal hunting has persisted because the prevalent drivers of illegal hunting in the Luangwa Valley are critical needs, people’s survival and sustaining livelihoods, and that these drivers of illegal hunting have not been addressed effectively. This study also provides a new understanding of how the prevalent drivers of illegal hunting and other factors such as defiance, the beliefs of local illegal hunters and the limitations of law enforcement may have operated in influencing the levels and persistence of illegal hunting in the Luangwa Valley.

### 4.1. Reliability, Validity and Trustworthiness of Study Instruments and Process

The reliability and validity of the questionnaire were deemed acceptable because the pre-tested questionnaire items showed that only few items had problems (which were corrected before data collection), most items were not wrongly responded to, responses were not mutually contradictory and the coefficient of Cronbach’s Alpha was 0.9, n = 86 (after the study’s data collection the coefficient was 0.8, n = 68). Heale and Twycross [53] indicated that the reliability of the questionnaire is considered acceptable when the coefficient of Cronbach’s Alpha is at least 0.7. Additionally, the qualitative data’s collection, analysis and interpretation were considered trustworthy based on the congruence and confirmability of the data among the FGDs and between FGDs and IDIs and the long period (over six months) of engagement with participants as recommended by Lincoln and Guba [54]. Furthermore, reporting and interpreting negative-case analyses and the recorded proof of evidence of the data collected in this study also enhanced the credibility and dependability of the data collected, data analysis and interpretation as guided by Guba and Lincoln [56], Cope [57] and Nowell et al. [58].

### 4.2. The Drivers, Intervention Measures and Persistence of Illegal Hunting

The study here provides the first comprehensive list of the drivers of illegal hunting, conceptualised as proximate, underlying and thematic, in the Luangwa Valley landscape. A total of 27 drivers of illegal hunting were identified by quantitative and qualitative approaches (see Table 2 and Appendix A). The quantitative survey approach determined 23 drivers of illegal hunting, whereas the qualitative approach identified 18 drivers. Four of the eighteen drivers of illegal hunting were exclusively identified through qualitative methods and included the behavioural intention to hunt illegally, the non-ownership of wildlife by communities, human encroachment and development and poor partnerships/collaborations. Most of the drivers of illegal hunting identified by the quantitative survey approach were similar to those identified in a scoping review study in Africa by Zyambo et al. [14], although this had fewer drivers (17 in total). The quantitative approach of this study had more identified drivers of illegal hunting probably due to the survey method, which included reformed local hunters and three other stakeholders as the sample population, which broadened the perspective, whereas the scoping review study in Africa only considered a sample population of local hunters. The study also identified slightly over double the number of drivers of illegal hunting determined in another study in Africa by Lindsey et al. [16]. Furthermore, this study identified more drivers of illegal hunting than earlier studies in the Luangwa Valley [32,33,34,39,40,41], implying that the current study identified the most comprehensive drivers of illegal hunting in the Luangwa Valley.

Here, the study provides supporting evidence for the hypothesis that the persistence of illegal hunting is associated with the prevalent drivers of illegal hunting in the Luangwa Valley. Among the eight most prevalent drivers of illegal hunting identified in the Luangwa Valley, five of these were categorised under a thematic driver—the need for survival and sustaining livelihoods (see Table 2). Despite being under a different thematic category, preventative killing and human–wildlife conflicts, as drivers of illegal hunting, also related directly to the people’s need for survival and sustaining livelihoods. This implies that seven of the eight prevalent drivers of illegal hunting related to people’s needs for survival and sustaining livelihoods. The prevalence of five illegal hunting drivers that related to people’s continual needs for survival and sustaining livelihoods creates a continuous demand for the illegal extraction of wildlife. This is because survival and sustaining livelihoods are critical human motivations, or needs, based on Maslow’s hierarchy of human needs and evolutionary pressure for successful reproduction and survival [61]. Further, among the twelve most prevalent drivers of illegal hunting, six were significantly associated with the persistence of illegal hunting and five of them were related to people’s need for survival and sustaining livelihoods. Thus, the prevalence of the drivers of illegal hunting and their significant association with the persistence of illegal hunting provide evidence that the persistence of illegal hunting is linked to people’s prevalent need for survival and sustaining livelihoods. The findings of this study are consistent with what other studies conducted elsewhere in Africa have found; illegal hunting is used as a strategy for survival and supporting livelihoods [40,62,63,64,65].

This study also found supporting evidence for the hypothesis that the persistence of illegal hunting is associated with prevalent drivers that were unsatisfactorily addressed by intervention measures. Firstly, the prevalence of the drivers of illegal hunting was not significantly correlated with the prevalence of their respective intervention measures (*r_s_* = −0.24, *df* = 9, *p* = 0.485). Ideally, there should be a significant positive correlation between the prevalence of drivers of illegal hunting and the prevalence of their respective intervention measures to increase the likelihood of addressing the drivers effectively. However, this study found a non-significant negative correlation which implied that the prevalence of the intervention measures did not commensurate or match with the prevalence of the drivers of illegal hunting in the Luangwa Valley. Secondly, the prevalence of drivers of illegal hunting was negatively correlated with the prevalence of the respective intervention measures with the most satisfactory performances (*r_s_* = −0.81, *df* = 9, *p* = 0.003) and positively correlated with the prevalence of the respective intervention measures with unsatisfactory performances (*r_s_* = 0.62, *df* = 9, *p* = 0.040). These denote that the intervention measures with the most satisfactory performances addressed less prevalent drivers of illegal hunting, whereas those with unsatisfactory performances dealt with more prevalent drivers. Thirdly, the persistence of illegal hunting was significantly associated with five of the seven intervention measures with unsatisfactory performances in addressing the drivers of illegal hunting that directly related to people’s need for survival and sustaining livelihoods. This suggests that the intervention measures did not match with nor satisfactorily address the most prevalent drivers of illegal hunting (the need for people’s survival and sustaining livelihoods) in the Luangwa Valley. Therefore, persistent illegal hunting in the Luangwa Valley was mainly driven by the prevalence of people’s need for survival and sustaining livelihoods, which was not effectively addressed. Considering that the need for survival and sustaining livelihoods is the most critical, communities may engage in or access whatever resource is readily available to survive and sustain their livelihoods. This implies that when the prevalent drivers of illegal hunting relate to people’s critical need for survival and sustaining livelihoods, and when these needs are not met, then the illegal hunting of wildlife is likely to be high, pervasive and persistent. Furthermore, any illegal hunting intervention strategy that does not address the poachers’ critical motivation for poaching is likely to be ineffective and thereby unsustainable in tackling the illegal hunting problem.

### 4.3. Defiance/Protesting Unfairness

Defiance or protesting unfairness, as a driver of illegal hunting, is defined based on the Defiance Theory (DT) and its prediction that environmental harm, which includes illegal hunting, will increase (or persist) as the legitimacy of conservation policies, tactics and authority decline [24]. Therefore, defiance usually occurs when local community members protest perceived unfairness or injustices by engaging in illegal hunting. The quantitative survey in this study identified defiance as one of the least prevalent drivers of illegal hunting (n = 10, 2.9%) in the Luangwa Valley. However, in the qualitative method of the FGDs, defiance was found to be one of the most prominent subthemes, with 19 references cited from five FGDs. The structured questionnaire in the quantitative survey probably restricted respondents from expressing their deeply rooted feelings and experiences related to defiance, whereas participants in the FGDs were able to express their resentment towards some injustices or unfairness caused by wildlife management regulations and practises. Thus, the results from the qualitative data analysis clarified and provided a deeper understanding of the significance of defiance as one of the drivers of illegal hunting in the FGDs. Importantly, participants in the FGDs provided six reasons (as earlier reported herein) for defiance which were remarkably related to people’s needs for survival and sustainable livelihoods in their communities. The prominence and credence of defiance were supported because the reasons for poaching wildlife in protest of unfairness were directly related to the prevalent proximate drivers of illegal hunting in the Luangwa Valley. This suggests that local communities are likely to protest any unfairness or injustices by hunting illegally when the perceived unfairness or injustices are related to the critical needs of the local communities that motivate them to poach wildlife. Therefore, defiance or protesting unfairness as a driver of illegal hunting augments the premise that illegal hunting has persisted in the Luangwa Valley mainly because of the prevalence of people’s critical need for survival and sustaining livelihoods, which has not been effectively addressed. This represents the first-time defiance or protesting unfairness is empirically identified as one of the drivers of illegal hunting in the Luangwa Valley landscape.

### 4.4. Beliefs and Behavioural Intentions to Hunt Illegally

Participants in the mostly Reformed Illegal Hunters FGDs expressed behavioural, normative and control beliefs about wildlife, illegal hunting and its benefits in 64 references cited from seven FGDs. The essence of the participants’ beliefs was that wildlife was given by God for people’s survival and that illegal hunting was good and an important means for helping suffering people who have no alternative options, which thereby made it difficult to stop poaching. The articulated beliefs of the participants were linked to their behavioural intentions to hunt illegally because these behavioural intentions were expressed in a consequential manner to these beliefs during FGDs. According to the Theory of Planned Behaviour (TPB), behavioural, subjective norms (normative) and perceived control beliefs determine both intention and behaviour [26,27], and behavioural intention is the most immediate determinant of social behaviour [27,66]. This suggests that a behavioural intention to hunt illegally is the most proximate driver of illegal hunting behaviour. Thus, the behavioural intention to hunt illegally energises other drivers of illegal hunting in influencing or mediating illegal hunting behaviour in an individual. The concept of behavioural intention has been used to assess behaviour change interventions and study potential predictors of illegal hunting [67,68]. Further, studies on behaviour change interventions and predicting behavioural intentions provide empirical clues that behavioural intentions to hunt illegally or to conserve are critical factors that determine whether people in a community who may be affected by similar drivers of illegal hunting end up hunting illegally or refraining from poaching, respectively [67,68]. Therefore, the behavioural intention to hunt illegally is considered the most immediate driver of illegal hunting behaviour, as argued by Zyambo et al. [14] in their conceptual framework on how underlying, proximate and the most proximate drivers may sequentially influence illegal hunting behaviour. The prominence of the theme behavioural intentions to hunt illegally, with 17 references in six FGDs, suggests that it could be pervasive among local illegal hunters and indicative of inadequate intervention measures in the area, which strengthen the beliefs of local communities that support conservation and weaken beliefs that encourage illegal hunting behaviour. This study represents the first time the behavioural intention to hunt illegally has been investigated and described as a driver of illegal hunting at the landscape level in the Luangwa Valley.

### 4.5. Limitations of Law Enforcement in Addressing Illegal Hunting

Despite being the most prevalent intervention measure (n = 213, 61.6%) in the Luangwa Valley, improved law enforcement addressed one of the least prevalent drivers of illegal hunting, weak/inadequate law enforcement (n = 39, 11.3%). The priority of improving law enforcement emanated from the perspective of Wildlife Agency Staff, which was that inadequate law enforcement was the most important driver of illegal hunting in the Luangwa Valley, as shown by its high prominence during the Wildlife Agency Staff FGD and the IDIs of conservation experts (see Appendix A). However, this study found evidence that inadequate law enforcement was not the major motivation for illegal hunting among local illegal hunters and that law enforcement had limitations in addressing illegal hunting in the Luangwa Valley. Although law enforcement was the most prioritised intervention measure and had the most satisfactory performance rating, this study found that the illegal hunting levels in the Luangwa Valley were moderate to high and had persisted for up to over 30 years. The results from the FGDs and IDIs indicated that the theme of the limitations of law enforcement was prominent, with a total of 24 references cited from six FGDs and three IDIs. Further, the results from the FGDs and IDIs also showed that the major limitation of law enforcement was that it did not effectively deter local illegal hunters from poaching wildlife mostly because local illegal hunters were mainly motivated by prevalent drivers like poverty and the need for income, bushmeat and a livelihood and not by weak law enforcement. This is consistent with what Milner-Gulland and Leader Williams [39] suggested in a study in the Luangwa Valley, that very high law enforcement was more unlikely to be effective in deterring local illegal hunters than non-local illegal hunters and that livelihood programmes were more successful deterrent measures for local illegal hunters. Likewise, Marks [30] reported that when law enforcement increased in the central Luangwa Valley, local hunters were not deterred from illegal hunting but changed their hunting methods from using firearms to snaring to avoid detection by law enforcement staff. Therefore, improved law enforcement may have had little or no effect on deterring local illegal hunters, who could be in larger numbers than non-local hunters, as suggested by the numbers of references cited on types of illegal hunters during the FGDs and IDIs (see Appendix A).

This implies that when law enforcement is the major intervention measure against illegal hunting that is mainly driven by the critical need for survival and sustaining livelihoods, then illegal hunting by local hunters is likely to be high, pervasive and persistent. This underscores the importance of identifying the drivers of illegal hunting and then targeting them with relevant intervention measures, instead of assuming intervention measures for dealing with poaching activities of unidentified motivations. However, improved law enforcement could be more effective in deterring non-local illegal hunters, as suggested by Milner-Gulland and Leader-Williams [39], probably because it increases the cost of illegal hunting for them, especially as they travel long distances to usually vast and less familiar terrains to poach wildlife. The increased cost of illegal hunting for non-local illegal hunters includes increased risks of being detected, increased risks of being arrested and increased risks of failing to achieve the objectives of their long-distance illegal hunting excursion.

### 4.6. Different Perspectives on Drivers of Illegal Hunting and Intervention Measures

The results from the quantitative data analysis showed that the perspectives of stakeholders on the drivers of illegal hunting and their intervention measures were different based on the significant differences found in the proportional means and distribution patterns of the responses from stakeholders in different strata. Firstly, the distribution of responses from Reformed Illegal Hunters was significantly more widespread than expected in identifying the most prevalent proximate drivers of illegal hunting that related to people’s need for survival and sustaining livelihoods. Secondly, the distribution of responses from Wildlife Agency Staff and Conservation-Interested Entities were significantly more widespread than expected in identifying the main underlying drivers of illegal hunting that were mostly not related to people’s need for survival and sustaining livelihoods. Thirdly, the response distribution patterns of the Reformed Illegal Hunters were significantly different from those of the Wildlife Agency Staff and Conservation-Interested Entities. The propensity for identifying proximate drivers of illegal hunting was higher among Reformed Illegal Hunters, whereas the propensity for identifying underlying drivers of illegal hunting was higher among Wildlife Agency Staff and Conservation-Interested Entities.

The qualitative data analysis in this study also highlighted the differences in their perspectives on the drivers of illegal hunting. The Wildlife Agency Staff FGD and three IDIs provided a total of 23 references that cited weak/inadequate law enforcement as the most prominent driver of illegal hunting. However, the Wildlife Agency Staff FGD did not contribute to the references that cited the most prominent drivers of illegal hunting, such as the need for income, poverty, human–wildlife conflicts and a lack of alternative livelihoods. The Wildlife Agency Staff and Conservation-Interested Entities considered weak law enforcement as the most important driver of illegal hunting and prioritised the improvement of law enforcement, and hence it was the most prevalent intervention measure implemented, even though weak law enforcement was among the least prevalent drivers of illegal hunting in the Luangwa Valley.

Hence, the perspectives of the resource users (Reformed Illegal Hunters) in the Luangwa Valley were more inclined toward the proximate drivers of illegal hunting that concerned people’s critical need for survival and sustaining livelihoods. However, the perspectives of resource managers (represented by Wildlife Agency Staff and Conservation-Interested Entities) were more disposed to underlying drivers of illegal hunting that did not directly deal with people’s need for survival and sustaining livelihoods, such as weak law enforcement, the high market demand for wildlife products, the lack of conservation education/awareness and human population increase/influx. This is consistent with what a study in Tanzania on factors contributing to illegal hunting found; resource managers’ perspectives concentrated on facilitating factors such inadequate patrol resources and impassable roads, whereas those of resource users focused on motivating factors such as limited income-generating opportunities and facilitating factors [69]. However, most studies on the drivers of illegal hunting in Africa have not compared the perspectives of resource users and resource managers. Remarkably, the perspectives of Reformed Illegal Hunters (direct wildlife resource users) on the drivers of illegal hunting also differed from those of Community Resource Boards, who are local community members. Thus, the perspectives of community members in general may not adequately represent those of direct resource users (hunters) on what motivates hunters to engage in illegal hunting. Furthermore, this implies that when the perspectives on the drivers of illegal hunting of resource users and managers are different, and resources managers do not consider the perspectives of resource users, it is most likely that anti-poaching strategies suggested by resource managers would be biased and inadequate for the holistic tackling of the poaching problem and might consequently lead to persistent illegal hunting. This underscores the importance of understanding and considering the perspectives of both resource users and managers on the drivers of the illegal extraction of natural resources when designing holistic and effective intervention strategies.

### 4.7. Proposed Postulation on the Persistence of Illegal Hunting

Based on the evidence provided in this study, we postulate that, in communities that surround or are adjacent to protected areas in the Luangwa Valley and elsewhere, (persistent) illegal hunting behaviour is a result of interactions of behavioural intentions to hunt illegally due to local beliefs (behavioural, normative and perceived control) with prevalent drivers of illegal hunting and motivations for defiance. This postulation takes into consideration possible dynamics in the prevalent drivers of illegal hunting and motivations for defiance and is based on the Theory of Planned Behaviour [25,26,27,65] and Defiance Theory [7,23,24]. Further, if there are prevalent drivers of illegal hunting in an area this implies that the intervention measures addressing them are ineffective. In this study, behavioural intentions to hunt illegally were influenced by beliefs (behavioural, normative and control) that were related to the prevalent drivers of illegal hunting and motivations for defiance.

### 4.8. Limitations of the Study

The evidence provided in this study is based on the associations between and correlations of variables, which may not imply causation in their relationships. However, these associations may be due to direct or indirect causations [70] which were not determined in this study. The variables in this study were numerous and there was a possibility of there being confounding variables and spurious associations. Furthermore, variables for associations and correlations in this study were largely based on views or perceptions, which are prone to perception bias. Therefore, this study may not have provided proof of direct or indirect causation but provided evidence of the most likely relationships, which are highlighted and supported by the findings of both the quantitative and qualitative study approaches used in this study and other studies conducted elsewhere in Africa.

### 4.9. Future Directions

We recommend that experimental studies be conducted to establish the causation of the associations of drivers of illegal hunting and intervention measures with the persistence of illegal hunting. This will provide proof of either the direct or indirect causations of persistent illegal hunting in the Luangwa Valley. Secondly, studies should be conducted in other landscapes to validate the postulation suggested in this study that, in communities that surround or are adjacent to protected areas, (persistent) illegal hunting behaviour is the result of the interaction of behavioural intentions to hunt illegally due to local beliefs with prevalent drivers of illegal hunting and motivations for defiance. Furthermore, based on the findings of this study, we propose the following new guidelines for addressing persistent illegal hunting in the Luangwa Valley: (i) design intervention measures for addressing the drivers of illegal hunting instead of targeting symptomatic illegal hunting activities, (ii) prioritise addressing key prevalent drivers of illegal hunting which relate to people’s critical need for survival and sustaining livelihoods, (iii) address the local beliefs that influence behavioural intentions to hunt illegally, (iv) address motivations for the defiance which is expressed by local people protesting perceived injustices or unfairness by hunting illegally and (v) improve and sustain law enforcement to deter non-local illegal hunters. We suggest that these proposed guidelines for addressing the drivers of illegal hunting may adaptively be applied in tackling the illegal harvesting of other natural resources in protected areas which are surrounded by local communities with similar socio-economic contexts.

## 5. Conclusions

Decades of persistent illegal hunting have resulted in severe wildlife population declines, with evidence of extirpation of the black rhinoceros, an endangered species in the Luangwa Valley. Despite various intervention measures, the problem of illegal hunting persisted and the reason for its persistence had not been clearly established. We analysed the drivers of illegal hunting and their intervention measures using quantitative and qualitative approaches to provide a deeper understanding of why illegal hunting has persisted in the Luangwa Valley. We have empirically established that persistent illegal hunting is mostly driven by the prevalence of people’s critical need for survival and sustaining livelihoods, which has not been effectively addressed, and that weak or inadequate law enforcement is not the main motivating factor as presumed by resource managers. In this study, we found the following problems that could be contributing to persistent illegal hunting of wildlife in the Luangwa Valley: (i) the prevalent drivers of illegal hunting in the area were related to people’s critical need for survival and sustaining livelihoods, (ii) the perspective of resource managers that inadequate law enforcement was the main driver of the illegal hunting differed from that of direct resource users, (iii) prevalent intervention measures did not commensurate with the prevalent drivers of illegal hunting, (iv) intervention measures with unsatisfactory performance ratings addressed the more prevalent drivers of illegal hunting, (v) prevalent law enforcement was ineffective in deterring local illegal hunters from poaching, because it did not address the critical drivers of illegal hunting, (vi) the behavioural intentions of local hunters to hunt illegally were pervasive in the landscape due to their beliefs about wildlife, illegal hunting and its benefits and (vii) local illegal hunters also poached wildlife in sheer defiance in order to protest perceived unfairness or injustices allowed by authorities. These are empirically generated insights and focus areas for addressing the problem of persistent illegal hunting in the landscape. Therefore, this study has contributed to the conceptual knowledge on how persistent illegal hunting may occur and, hence, to our practical understanding of what could be helpful in addressing the poaching problem in African landscapes with protected areas that are surrounded by local communities. The profound implication of these findings is that where the illegal harvesting of natural resources in protected areas by local resource users is driven by the critical need for survival and livelihoods, which are not effectively addressed, illegal harvesting may persist even with increased law enforcement. We hope this novel knowledge and understanding of persistent illegal hunting in the Luangwa Valley will provide valuable information for researchers, policy makers and resource managers to increase their understanding and knowledge base, specific and sustainable policy directions, and effective wildlife resource management. However, if the status quo is maintained, then illegal hunting will persist in the Luangwa Valley and continue to undermine wildlife and biodiversity conservation, tourism development and sustainable community livelihoods.

## Figures and Tables

**Figure 1 animals-14-02401-f001:**
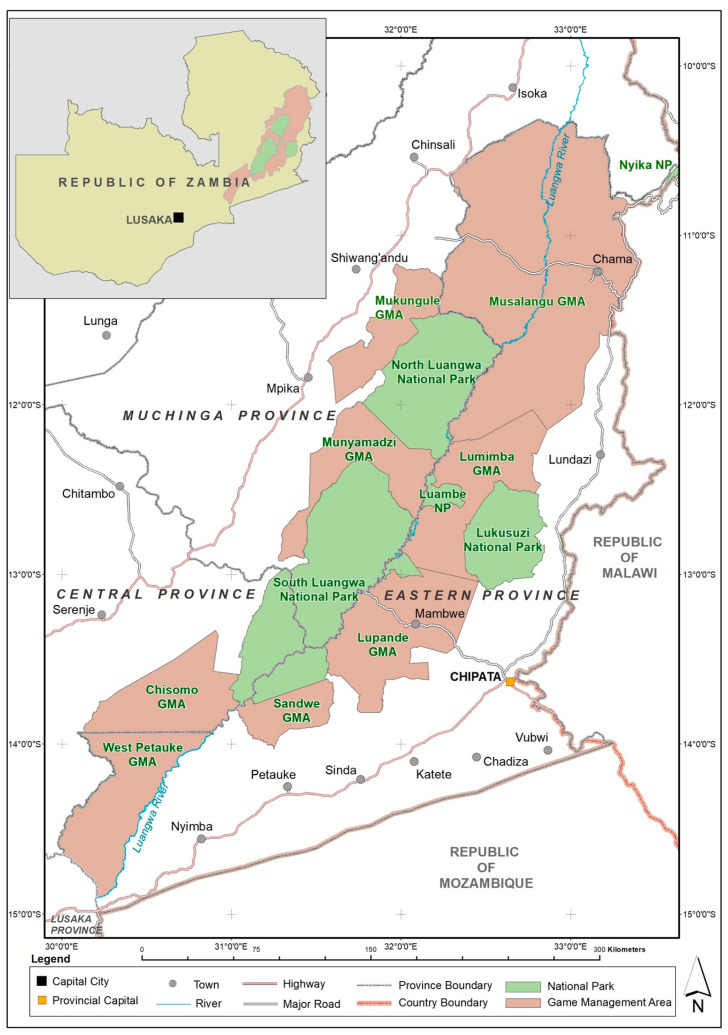
Geographical location of the study area comprising eight Game Management Areas (GMAs) that are adjacent to four National Parks in the Luangwa Valley.

**Figure 2 animals-14-02401-f002:**
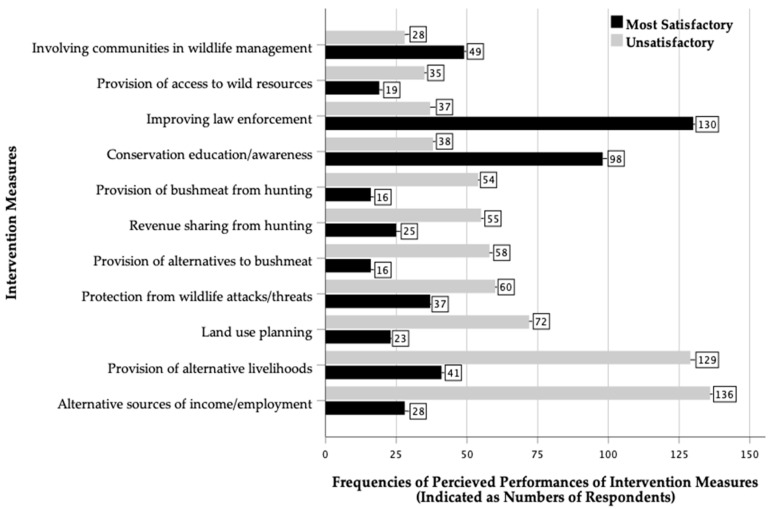
Frequencies of the perceived performances of intervention measures in addressing the drivers of illegal hunting identified in a questionnaire survey in the Luangwa Valley.

**Table 1 animals-14-02401-t001:** Number of responses on the levels, persistence and trends in illegal hunting to a questionnaire survey in the Luangwa Valley.

Sampling Strata	Illegal Hunting Status in the Luangwa Valley									
	Levels			Persistence				Trends		
	Low	Moderate	High	<1 Year (Starting)	1–14 Years (Short)	15–30 Years (Long)	>30 Years (Very Long)	Decreased	Stable	Increased
Reformed Illegal Hunters	93	42	7	0	108	30	4	93	27	21
Community Resource Boards	40	20	3	1	32	11	18	49	13	1
Wildlife Agency Staff	9	55	30	5	33	26	30	48	29	17
Conservation-Interested Entities	13	22	10	0	11	12	22	20	16	9
**Total**	155 (44.8%)	139 (40.2%)	50 (14.5%)	6 (1.7%)	184 (53.2%)	79 (22.8%)	74 (21.4%)	210 (60.7%)	85 (24.6%)	48 (13.9%)

**Table 2 animals-14-02401-t002:** Frequencies and categories of drivers of illegal hunting identified in a questionnaire survey in the Luangwa Valley.

* Drivers of Illegal Hunting	No. of Respondents Identifying Drivers (% in Parentheses)	Proximate/ Underlying Drivers	Thematic Drivers
Lack of alternative sources of income/employment	197 (56.9%)	Proximate	Need for survival and sustaining livelihoods
Poverty	195 (56.4%)	Underlying	Need for survival and sustaining livelihoods
Need for bushmeat consumption	183 (52.9%)	Proximate	Need for survival and sustaining livelihoods
Need for income from bushmeat and animal products	180 (52.0%)	Proximate	Need for survival and sustaining livelihoods
Sponsorship to hunt illegally	132 (38.2%)	Underlying	External/internal sponsorship
Lack of sources of meat/ protein	112 (32.4%)	Proximate	Need for survival and sustaining livelihoods
Retaliatory killing	96 (27.7%)	Proximate	Human–wildlife conflicts
Preventative killing	84 (24.3%)	Proximate	Human–wildlife conflicts
Human–wildlife conflicts	79 (22.9%)	Underlying	Human–wildlife conflicts
Demand for wildlife products	70 (20.2%)	Underlying	Market demand for wildlife products
Lack of/inadequate conservation education/awareness	70 (20.2%)	Underlying	Lack of conservation education/awareness
Lack of/inadequate tangible benefits from conservation	52 (15.0%)	Underlying	Need for survival and sustaining livelihoods
Population influx/increase	49 (14.2%)	Underlying	Demographic growth
Inadequate community involvement in wildlife management	47 (13.6%)	Underlying	Inadequate devolution of wildlife management
Weak/inadequate law enforcement	39 (11.3%)	Underlying	Inadequate legislation/ enforcement
Need for trophies for income/use	34 (9.8%)	Proximate	Need for survival and sustaining livelihoods
Cultural/traditional needs	20 (5.9%)	Proximate	Cultural needs/ significance
Political influence	15 (4.3%)	Underlying	Political influence
Defiance/protest	10 (2.9%)	Proximate	Defiance/protesting unfairness
Recreational/sports needs	5 (1.4%)	Proximate	Recreational need
Desire to outsmart law enforcement staff	5 (1.4%)	Proximate	Desire to outsmart law enforcement staff

* Drivers of illegal hunting identified by less than five respondents such as the need to practise shooting with a firearm (n = 2, 0.6%) and a desire for the pleasure in killing animals (n = 2, 0.6%) are not included in the table.

**Table 3 animals-14-02401-t003:** Prevalence of intervention measures implemented to address the drivers of illegal hunting identified in a questionnaire survey in the Luangwa Valley.

Intervention Measures	No. of Respondents Identifying Intervention Measures	Percent (%)
Improving law enforcement	213	61.6
Providing conservation education/awareness	207	59.8
Provision of alternative livelihoods	187	54.0
Provision of alternative sources of income/ employment	152	43.9
Involving communities in wildlife management	112	32.4
Protecting communities from animal attacks and threats	99	28.6
Revenue sharing from hunting	80	23.1
Land use planning	62	17.9
Provision of bushmeat from hunting	49	14.2
Provision of alternative to bushmeat	27	7.8
Provision of access to wild resources	26	7.5

**Table 4 animals-14-02401-t004:** The association between the persistence of illegal hunting and the drivers of illegal hunting that relate to people’s need for survival and sustaining livelihoods in the Luangwa Valley.

Driver of Illegal Hunting	* Likelihood Ratio	Degrees of Freedom (df)	Cramer’s V	*p*-Value	Decision	Comments
Need for bushmeat consumption	23.209	3	0.243 (*p* < 0.001)	<0.001	Reject null hypothesis	Evidence of a moderate association
Need for income from bushmeat	8.019	3	0.152 (*p* = 0.047)	=0.046	Reject null hypothesis	Evidence of a weak association
Preventative killing	16.626	3	0.200 (*p* = 0.003)	<0.001	Reject null hypothesis	Evidence of a moderate association
Human–wildlife conflicts	20.129	3	0.243 (*p* < 0.001)	<0.001	Reject null hypothesis	Evidence of a moderate association
Need for trophies for income/use	13.745	3	0.206 (*p* = 0.002)	=0.003	Reject null hypothesis	Evidence of a moderate association
Lack of tangible benefits from conservation	14.296	3	0.202 (*p* < 0.001)	=0.003	Reject null hypothesis	Evidence of a moderate association
Poverty	2.651	3	0.087 (*p* = 0.451)	=0.449	Retain null hypothesis	No evidence of association
Lack of alternative income/employment	3.358	3	0.098 (*p* = 0.347)	=0.340	Retain null hypothesis	No evidence of association
Lack of alternative source of meat	3.374	3	0.096 (*p* = 0.360)	=0.338	Retain null hypothesis	No evidence of association
Retaliatory killing	5.154	3	0.161 (*p* = 0.169)	=0.169	Retain null hypothesis	No evidence of association

* The likelihood ratios were considered for testing associations instead of the Pearson Chi-Square values as some cells of the expected counts were less than five (5).

**Table 5 animals-14-02401-t005:** The association between the persistence of illegal hunting and the unsatisfactory performance of intervention measures in addressing the drivers of illegal hunting that relate to people’s need for survival and sustaining livelihoods in the Luangwa Valley.

Intervention Measures with Unsatisfactory Performance	* Likelihood Ratio	Degrees of Freedom (df)	Cramer’s V	*p*-Value	Decision	Comments
Provision of alternative livelihoods	13.367	3	0.253 (*p* = 0.004)	=0.004	Reject null hypothesis	Evidence for a moderate association
Provision of alternatives to bushmeat	32.488	3	0.366 (*p* < 0.001)	<0.001	Reject null hypothesis	Evidence for a strong association
Provision of bushmeat from hunting	19.029	3	0.276 (*p* < 0.001)	<0.001	Reject null hypothesis	Evidence for a moderate association
Revenue sharing from hunting	34.533	3	0.372 (*p* < 0.001)	<0.001	Reject null hypothesis	Evidence for a strong association
Provision of access to wild resources	11.980	3	0.305 (*p* = 0.01)	=0.007	Reject null hypothesis	Evidence for a strong association
Provision of alternative employment/income	5.476	3	0.156 (*p* = 0.136)	=0.140	Retain null hypothesis	No evidence for association
Protection of communities from attacks and threats from wild animals	0.122	3	0.024 (*p* = 0.989)	=0.989	Retain null hypothesis	No evidence for association

* The likelihood ratios were considered for testing associations instead of the Pearson Chi-Square values as some cells of the expected counts were less than five (5).

## Data Availability

The data presented in this study are available on request from the corresponding author as the data are still being used in the PhD studies of the first author of this article. This is to ensure that data critical to the PhD study are not published by others before the study is completed.

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
