# Peer review of "Persistent Illegal Hunting of Wildlife in an African Landscape: Insights from a Study in the Luangwa Valley, Zambia"

_animals, 2024, doi:10.3390/ani14162401_

Round 1

Reviewer 1 Report

Comments and Suggestions for Authors

Manuscript animals-3056225–V1

This is an interesting study that investigates drivers of illegal hunting and intervention measures using quantitative and qualitative approaches to better understand why illegal hunting persisted among local communities in the Luangwa Valley (Zambia), an important issue with crucial social, ecological and conservation implications.

However, I have some comments and suggestions that should be addressed and hop will help improve the manuscript. Moreover, despite the great potential of the work, which I am sure will be of great utility in improving the management of illegal hunting by policy makers and resource managers, my main concern lies in the excessive length of some sections, especially of the results and the discussion. In the case of the results, although well written and clearly reflecting the main findings of the work, I think they should be reduced to about 50%. Therefore, authors should make a significant effort to synthesize the aforementioned sections, merging some sentences/paragraphs with the Materials and Methods (Data Analyses) and the Discussion sections, avoiding considerable volume of redundant information especially in the Results (see some examples below), or even moving some parts of the main text to the Supplementary material. I believe that in this way the manuscript will present a greater flow that will substantially improve the understanding of the main findings and enhance its potential.

Specific comments

Lines 61–63. See also Margalida and Mateo 2019 Science to support this statement.

Lines 99–105. I think it is also important to mention the effect of illegal hunting of other non-mammalian species, such as vultures (a crucial functional group that provides essential ecosystem services), because of the serious conservation problem of their direct and indirect persecution, particularly in Africa. See for example the works of Ogada et al. 2016 Conserv. Lett., Santangeli et al. 2016 Biol. Conserv., Margalida et al. 2019 Science, Safford et al. 2019 Conserv. Int.

Lines 135–139: Suggest rewriting the sentence as follows: “Therefore, the first hypothesis (H1) is: persistent….” and “The second hypothesis (H2) is: persistent….”

Line 172: Suggest adding a coma after (CRBs).

Lines 181–185: Suggest enumerating the four strata: 1) Reformed Illegal Hunters, 2) Community Resource Board members, 3) Wildlife Agency Staff and 4) staff members of Conservation Interested Entities, as has been done on lines 887–894.

Line 215: Suggest change “comprise” to “comprised”.

Line 223: Add a coma after country.

Line 224: Check the citation Agency et al., 2019 since it appears different in the reference list and is the only citation in which the year is mentioned in the main text of the manuscript.

Line 255: Add a coma after strata.

Line 493. Indicate ‘that’ congruence was attained sounds better?

Line 509. I think that in order to lighten the text of the results, the name of the four strata can be removed from this sentence as they are already mentioned previously in lines 181–185.

Line 521. Check the percentage, I believe it should be 50.3% (174/346), as it appears correctly on line 516.

Lines 526–558. Check all the percentages because some of them are not calculated on a total of 346 respondents, but seem to be calculated on 344 and 343 (as can be deduced from Table 1). If this is the case, it should be explained in the text.

Line 541. Suggest removing the number inside the parentheses here and throughout the manuscript.

Line 559: In summary, these results…

Lines 599–608: Here the authors should reference Figure 2, and not only cite it at the beginning of the section in line 567. Similarly, all tables should also be referenced in each sentence when discussing the results that appear in them, and not only reference them at the beginning of the different subsections.

Lines 611–613: Add space before and after the = symbol, as has been done throughout the manuscript.

Line 648–650: “The prevalence of drivers of illegal hunting and the prevalence of respective intervention measures and intervention performances were tested to ascertain their relationships using Spearman’s rank correlation coefficient”. I suggest removing this paragraph from the results section as this information already appears in section 2.6 Data Analyses, which is where all statistical treatments performed should be detailed. Similarly, I suggest removing lines 725–734 from the this section and explain the hypotheses tested only in section 2.6 Data Analyses (subsection 2.6.4 Hypotheses testing, as it is already done), as this is where such statistical details should be given. In this way, the results can be understood in a more direct, fluent and understandable way.

Lines 665–673: This paragraph seems out of place here. Authors should summarize the most relevant results in this section (Results) and avoid interpretations and sentences that need to be explained in the discussion. I believe that the entire results section should be synthesised in a substantial way and avoid discussing the results in this section, which I currently find, despite being well written, excessively long and with redundant information (e.g., lines 711–721, 769–775), which should be detailed in the discussion section.

Lines 677–678: Similar to my previous comments, I believe that this sentence should be removed from the results, as it represents redundant information that should be adequately explained in the section 2.6 Data Analyses (subsection 2.6.3, lines 407–409). Here, the paragraph should start with “The proportion means of responses in identifying...”.

Lines 736–738 and 755–758. To reduce the length of the results, I suggest avoiding mentioning the six drivers of illegal hunting in the main text and refer only to Table 4 where it can already be clearly seen which are the significant drivers. Similarly, I suggest avoiding detailing intervention measures in the text and refer only Table 5 in the sentence.

Line 854. Suggest mentioning the number of participants in written form (Nine) or in numerical value (9), but it is not necessary to do it both ways. Please check this here and throughout the manuscript (e.g., subsection 3.3.1, lines 945–946…).

Lines 856–858. This information can already be seen in Table 7 and it is not necessary to mention all the broad themes emerged from thematic data analysis, it would be sufficient to refer to Table 7.

Line 887. Suggest changing “;” to “:”.

Line 898–900. I believe that the examples provided by the participants in the open questions are very important and useful to better understand the topic addressed in this study and to interpret the results obtained in the qualitative and quantitative approach. However, perhaps only the most relevant ones could be mentioned in the main text and the rest could be moved to the supplementary material, as this information makes the results section somewhat longer. Moreover, some of them are repeated (e.g. lines 1018–1020 and 965–970), providing redundant information that make the results section too long and costly to follow.

Line 904. To reduce the length of the manuscript, consider moving Table 7 to the supplementary material or try to present these results by means of a figure, as it would be more attractive and visual.

Line 995. ‘Altitudes’ or ‘attitudes’? Please, check it.

Line 957. Please check the spelling throughout the manuscript, do you want US or UK spelling? The text is a mixture (e.g. behavior/behaviour, behavioral/behavioural, see lines 1184–1185).

Line 1031: Suggest removing all p-values indicating significant models from here and throughout the discussion (e.g., lines 1249, 1252…), this information should not be detailed in the discussion section as it is already explained in the Data Analyses section and in the results.

Line 1089: I think it is not necessary to repeat, at the end of many sentences, ‘in the Luangwa Valley’, as the focal area of study is already mentioned in the first sentence of the discussion. It is not necessary to remove it from all sentences, but from those where it is redundant, such as in line 1089-1090 (as it is mentioned in the previous line). Please check this throughout the discussion.

Lines 1271–1277: This is a very relevant finding of the present work. In addition to discussing the results with the findings found by Kisingo et al. 2022 in Tanzania, the authors should discuss and compare their findings with those observed in other regions of Africa and in other continents.

Line 1337–1339: Suggest moving “and” before (iv) to before (v).

Line 1346–1347: Suggest mentioning the species name, as follows: “Decades of persistent illegal hunting had resulted in severe wildlife population declines with evidence of extirpation of [the xxx], an endangered species in the Luangwa Valley”.

Comments on the Quality of English Language

The manuscript is generally well written, and there is clearly value in the work, which presents a valuable contribution to the field and provides a broader framework for identifying and understanding the drivers and persistence of poaching in the study area that can also be useful in other regions.

Author Response

For research article

Response to Reviewer X Comments

1. Summary

Thank you very much for taking the time to review this manuscript. Please find the detailed responses below and the corresponding revisions/corrections highlighted/in track changes in the re-submitted files. [This is only a recommended summary. Please feel free to adjust it. We do suggest maintaining a neutral tone and thanking the reviewers for their contribution although the comments may be negative or off-target. If you disagree with the reviewer's comments please include any concerns you may have in the letter to the Academic Editor.]

2. Questions for General Evaluation

Reviewer’s Evaluation

Response and Revisions

Does the introduction provide sufficient background and include all relevant references?

Yes/Can be improved/Must be improved/Not applicable

[Please give your response if necessary. Or you can also give your corresponding response in the point-by-point response letter. The same as below]

Are all the cited references relevant to the research?

Yes/Can be improved/Must be improved/Not applicable

Is the research design appropriate?

Yes/Can be improved/Must be improved/Not applicable

Are the methods adequately described?

Yes/Can be improved/Must be improved/Not applicable

Are the results clearly presented?

Yes/Can be improved/Must be improved/Not applicable

Are the conclusions supported by the results?

Yes/Can be improved/Must be improved/Not applicable

3. Point-by-point response to Comments and Suggestions for Authors

Comments 1: [Paste the full reviewer comment here.]

Response 1: [Type your response here and mark your revisions in red] Thank you for pointing this out. I/We agree with this comment. Therefore, I/we have.[Explain what change you have made. Mention exactly where in the revised manuscript this change can be found – page number, paragraph, and line.]

“[updated text in the manuscript if necessary]”

Comments 2: [Paste the full comment here.]

Response 2: Agree. I/We have, accordingly, done/revised/changed/modified…..to emphasize this point. Discuss the changes made, providing the necessary explanation/clarification. Mention exactly where in the revised manuscript this change can be found – page number, paragraph, and line.]

“[updated text in the manuscript if necessary]”

4. Response to Comments on the Quality of English Language

Point 1:

Response 1:    (in red)

5. Additional clarifications

[Here, mention any other clarifications you would like to provide to the journal editor/reviewer.]

Reviewer 2 Report

Comments and Suggestions for Authors

Initially, I would like to extend my compliments to the authors for their efforts in data collection through a meticulously designed questionnaire model aimed at elucidating the reasons why illegal hunting persists in the studied area, despite ongoing mitigation efforts. This work will enable the implementation of strategic actions to mitigate the identified causes, even though it involves significant challenges.

The manuscript was very well conceived, demonstrating the authors' clear expertise on the subject. I congratulate them on the interesting material, rich in details in the methodology and discussion, and extremely well written, with appropriate and correct English. This reviewer reports having had great satisfaction in reading the manuscript, despite its length and the concentration required due to the richness of details.

Only two minor points caught this reviewer's attention:

1. Lines 143 to 153 – This section does not belong in the introduction as it presents the study's findings. It should be moved to the Discussion section.

2. Objective – At the end of the introduction, it is expected that the last paragraph directly presents the study's objective. This reviewer noticed the absence of this information.

Considering the points above, this reviewer recommends the publication of this manuscript after these minor corrections. However, if the authors find it interesting to enrich the discussion, this reviewer provides the following additional comments. I emphasize that the inclusion or not of these comments is at the authors' discretion, and their exclusion will not alter this reviewer's decision to recommend the publication of this article.

A) During the reading of this manuscript, this reviewer deeply perceived how the concept of "One Conservation" (https://doi.org/10.1590/1984-3143-AR2021-0024) is extremely valid for the issue encountered. One of the points of this holistic conservation vision is the need to include society as a whole. The presented manuscript shows fantastic investigative work aimed at understanding the persistence of hunting in the studied region. This study highlights the importance of a multidisciplinary approach in research on persistent hunting, even with mitigation actions, and is crucial for developing effective and sustainable techniques for wildlife conservation in the studied region. This is fully aligned with the concept of "One Conservation". It may be worth considering this concept in the discussion.

B) The use of GPS collars is predominantly associated with the research of wild species, focusing on their behavior, migration patterns, habitat use, and conservation status. However, it is known that it can be a tool for mitigating poaching – or at least a tool for deepening the understanding of its causes. Could this be a useful tool for the study or even for inhibiting hunting in the studied region? Below are some articles on the topic:

- GPS collars as a tool to uncover environmental crimes: https://doi.org/10.1111/acv.12826

General on GPS/Poaching:

- https://doi.org/10.3390/s18051474 

- https://doi.org/10.1111/1365-2664.12452

Tigers:

- https://doi.org/10.1109/IAIM.2017.8402602

- https://files.cfc.umt.edu/heblab/CatNews_Miller%20Tiger%20Translocation%202011.pdf

Elephants:

- https://doi.org/10.1016/j.ecolind.2017.08.039

C) The issue of hunting for critical needs for survival and sustaining livelihoods is not unique to the studied region, nor even to the African continent. The same challenge occurs in South America. Below are some articles that address the issue on the South American continent and that might contribute to a more comprehensive discussion:

- https://doi.org/10.1126/sciadv.abo5774

- https://doi.org/10.1007/s10745-023-00408-x

- https://doi.org/10.1186/s13002-022-00570-4

- https://eprints.lancs.ac.uk/id/eprint/86323/

- https://doi.org/10.3389/fcosc.2023.1221206

- https://doi.org/10.17635/lancaster/thesis/21

Author Response

(The authors gave the same response as above.)

Round 2

Reviewer 1 Report

Comments and Suggestions for Authors

Authors have considered in detail all my comments and suggestions, and am happy with the improvement of the length and flow of the text and the overall scientific quality of the work. Clearly, authors have made an effort to reduce the length of the results and discussion sections that I pointed out before, and I have no further comments on this paper (only change "behavior" to "behaviour" in line 1234 of the new version).

I would like to congratulate the authors for this work, which well extends beyond its scientific importance and we clearly need more studies of its kind.